# The Three Two-Pore Channel Subtypes from Rabbit Exhibit Distinct Sensitivity to Phosphoinositides, Voltage, and Extracytosolic pH

**DOI:** 10.3390/cells11132006

**Published:** 2022-06-23

**Authors:** Xinghua Feng, Jian Xiong, Weijie Cai, Jin-Bin Tian, Michael X. Zhu

**Affiliations:** 1Department of Integrative Biology and Pharmacology, McGovern Medical School, The University of Texas Health Science Center at Houston, Houston, TX 77030, USA; lysoworker@zju.edu.cn (X.F.); xiongj@stanford.edu (J.X.); jin.bin.tian@uth.tmc.edu (J.-B.T.); 2Department of Cardiology, The Second Affiliated Hospital, Zhejiang University School of Medicine, Hangzhou 310000, China; 3Liangzhu Laboratory, Zhejiang University Medical Center, Hangzhou 311121, China; 4Program in Biochemistry and Cell Biology, MD Anderson Cancer Center and UTHealth Graduate School of Biomedical Sciences, Houston, TX 77030, USA; 5Zhejiang University Medical Center, Hangzhou 310000, China; weijiecai96@gmail.com

**Keywords:** electrophysiology, ion channel, endosome, lysosome, TPCN1, TPCN2, TPCN3, phosphoinositide, voltage dependence, sodium channels

## Abstract

Two pore channels (TPCs) are implicated in vesicle trafficking, virus infection, and autophagy regulation. As Na^+^- or Ca^2+^-permeable channels, TPCs have been reported to be activated by NAADP, PI(3,5)P_2_, and/or high voltage. However, a comparative study on the function and regulation of the three mammalian TPC subtypes is currently lacking. Here, we used the electrophysiological recording of enlarged endolysosome vacuoles, inside-out and outside-out membrane patches to examine the three TPCs of rabbit (*Oryctolagus cuniculus*, or *Oc*) heterologously expressed in HEK293 cells. While PI(3,5)P_2_ evoked Na^+^ currents with a potency order of *Oc*TPC1 > *Oc*TPC3 > *Oc*TPC2, only *Oc*TPC2 displayed a strict dependence on PI(3,5)P_2_. Both *Oc*TPC1 and *Oc*TPC3 were activatable by PI3P and *Oc*TPC3 was also activated by additional phosphoinositide species. While *Oc*TPC2 was voltage-independent, *Oc*TPC1 and *Oc*TPC3 showed voltage dependence with *Oc*TPC3 depending on high positive voltages. Finally, while *Oc*TPC2 preferred a luminal pH of 4.6–6.0 in endolysosomes, *Oc*TPC1 was strongly inhibited by extracytosolic pH 5.0 in both voltage-dependent and -independent manners, and *Oc*TPC3 was inhibited by pH 6.0 but potentiated by pH 8.0. Thus, the three *Oc*TPCs form phosphoinositide-activated Na^+^ channels with different ligand selectivity, voltage dependence, and extracytosolic pH sensitivity, which likely are optimally tuned for function in specific endolysosomal populations.

## 1. Introduction

Two-pore channels (TPCs) or two-pore segment channels are members of the superfamily of voltage-gated ion channels found in both plants and animals. Their membrane topology and structural architectures suggest that they represent evolutionary intermediaries between the single Shaker-like 6-transmembrane (TM) domain channels, including many K^+^ channels, TRP (transient receptor potential) channels, and cyclic nucleotide-gated channels, and the four-repeats 24-TM segment voltage-gated Na^+^ and Ca^2+^ channels [1,2,3]. Thus, each TPC protein contains 12-TM segments segregated in two homologous 6-TM repeats that are connected via a large cytoplasmic loop. Both the amino and carboxyl termini of the TPC also face the cytoplasmic side. The channel is formed by two TPC subunits as a dimer. High-resolution structures of *Arabidopsis* TPC1, mouse TPC1, human TPC2, and zebrafish TPC3 have been made available by X-ray crystallography and single-particle cryogenic electron microscopy (cryo-EM) [4,5,6,7,8], which, in addition to supporting the predicted topology and dimeric organization, have revealed novel insights on the gating mechanisms of TPCs by membrane voltage and the phospholipid agonist, phosphatidylinositol 3,5-bisphosphate [PI(3,5)P_2_] [6,7].

While most voltage-gated channels exert their ion-conducting function on the plasma membrane, TPCs are mainly known as organellar channels. In most plants, there is a single TPC gene, TPC1, which is found in plant vacuoles [9]. In animals, the number of TPC genes (also known as *TPCN* genes) ranges from zero to three, with most vertebrate species having three, TPC1, TPC2, and TPC3, except for humans, mice, and rats where TPC3 is missing [1,2,3,10]. Notably, despite the same membrane topology and structural organization, the sequence homology is low among TPC subtypes. Of note, *Arabidopsis* TPC1 is not an orthologue of animal TPC1 but rather equally distant from all three mammalian TPCs [1,2]. Mammalian TPCs are mainly localized in the membranes of endosomes and lysosomes, with TPC1 being preferentially in the endosomes and TPC2 in the lysosome [1,11,12,13]. TPC3 has been localized to recycling endosomes, lysosomes, and plasma membranes [10]. Thus far, the most prominent functional significance of mammalian TPCs has been their involvement in endolysosome trafficking, including regulation of autophagy flux and macropinocytosis [13,14,15], nutrient sensing regulated through the mammalian target of rapamycin (mTOR) [16], and cellular infection of virulent viruses, such as Ebola virus and SARS-CoV-2 [17,18]. The knockout of either *Tpcn1* or *Tpcn2* in mice results in endolysosomal trafficking defects, evidenced by endolysosomal accumulations of cholesterol, endocytosed growth factors, low-density lipoproteins, bacterial toxins, and viruses [12,17,19,20]. In addition, TPC2 also plays a role in exocytosis in various cell types [21]. 

For TPC3, however, the functional significance remains elusive. TPC3 from zebrafish and the clawed frog was reported to function as a voltage-gated non-inactivating Na^+^ channel on the plasma membrane, which is activated only at high positive potentials, producing ultra-long action potentials [22]. Although ultra-long action potentials have been reported in various species and different cell types, their precise roles are unclear [22]. In addition, it is unlikely that TPC3 underlies all forms of ultra-long action potentials, especially given the absence of this gene in primates and certain rodents [2,10]. Yet, when overexpressed in HEK293 cells, both rabbit TPC3 and chicken TPC3 showed intracellular localization in organelles, including recycling endosomes, late endosomes, and lysosomes, in addition to localization on the plasma membrane [11]. 

Although controversy exists, vertebrate TPCs are generally considered to be activated by nicotinic acid adenine dinucleotide phosphate (NAADP), PI(3,5)P_2_, or voltage. NAADP is thought to be the most potent Ca^2+^ mobilizing intracellular messenger that induces Ca^2+^ release from lysosome-like acidic organelles [23,24]. The initial characterization of cloned vertebrate TPC1-3, including those from mammals and sea urchins, all showed enhanced responses to NAADP-induced intracellular Ca^2+^ rise [1,11,25,26,27], with some exceptions on TPC3 from certain species [11,28]. However, the electrophysiological characterization of the expressed TPC1 and TPC2 using whole-endolysosome patch-clamp techniques revealed activation of mainly Na^+^-selective currents by PI(3,5)P_2_ [16,29,30]. Although in some studies NAADP also evoked Na^+^ and Ca^2+^ currents in whole-endolysosome patches containing TPC2 [31,32,33], neither the success rate nor the current amplitude elicited by NAADP matched that by PI(3,5)P_2_. Thus, either NAADP and PI(3,5)P_2_ activate these channels in different modes, representing distinct open conformations, or an auxiliary NAADP-binding protein is needed to reconstitute the full response of TPC1 or TPC2 to NAADP [21]. The former idea is supported by the recent identification of two structurally distinct synthetic agonists, TPC2-A1-N and TPC2-A1-P, that mimicked TPC2 activation by NAADP and PI(3,5)P_2_, respectively [34]; the latter is encouraged by the recent findings of two NAADP-binding proteins, JPT2 [35,36] and Lsm12 [37]. 

In inside-out patches excised from *Xenopus* oocytes, the heterologously expressed *Xenopus tropicalis* TPC3 was activatable by PI(3,4)P_2_, PI(3,5)P_2_, and PI(3,4,5)P_3_, but not PI(4,5)P_2_, when combined with a strong depolarization to a high positive voltage [38,39]. This contrasts with the study of zebrafish TPC3 heterologously expressed in HEK293 cells, which found the fish TPC3 to be insensitive to both PI(3,5)P_2_ and PI(4,5)P_2_ [22]. Furthermore, while the TPCs are thought to be evolution intermediaries between the single domain voltage-gated K^+^ channels and the four-domain voltage-gated Na^+^ and Ca^2+^ channels, voltage-dependent gating has mainly been found for TPC1 and TPC3 [22,30], with the voltage sensitivity of TPC2 being reportedly agonist dependent [40]. Thus, the three vertebrate TPCs not only share common features, e.g., activation by NAADP or PI(3,5)P_2_, but they are also quite different in terms of voltage sensitivity and subcellular localization. 

To define the similarities and differences among TPC isoforms, we sought to compare the three TPC subtypes from the same mammalian species. Here, we isolated cDNAs of TPC1 and TPC2 by RT-PCR from rabbit kidneys. Together with the previously cloned rabbit TPC3, we examined the responses of the three rabbit (*Oryctolagus cuniculus*, or *Oc*) TPC subtypes to phosphoinositides, voltage, and pH using voltage-clamp recordings of inside-out, outside-out, and whole-endolysosome patches from HEK293 cells that transiently expressed the individual *Oc*TPC subtypes. We show that all three *Oc*TPCs are sensitive to PI(3,5)P_2_ to conduct Na^+^ selective currents, but with marked differences in selectivity among other phosphoinositides, voltage dependence, and pH preference. These findings support the idea that the three mammalian TPCs play distinct functions likely in different acidic organelles and on the cell surface.

## 2. Materials and Methods

Reagents and cDNA constructs: All chemicals and reagents were obtained from Thermo Fisher (Waltham, MA, USA) or MilliporeSigma (Burlington, MA, USA) unless indicated otherwise. The cDNA for *Oc*TPC3 has been described before [11]. The cDNAs for *Oc*TPC1 and *Oc*TPC2 were obtained from rabbit kidneys using RT-PCR. Briefly, rabbit kidney total RNA (Zyagen, Inc, San Diego, CA, USA) was reverse transcribed using ReverTra Ace reverse transcriptase (TOYOBO USA., Inc., New York, NY, USA) and gene-specific primers (Appendix A). The cDNAs of *Oc*TPC1 and *Oc*TPC2 were obtained by PCR using Phusion DNA polymerase (New England Biolabs, Ipswich, MA, USA) and primers (Appendix A) designed based on GenBank sequences, GBCT01172891 and GBCT01193833, respectively. The PCR products were subcloned into pEGFP-N1 vector (Clontech, Mountain View, CA, USA) for expression as EGFP fusion proteins. *Oc*TPC1-R540I mutation was made by PCR using *Oc*TPC1-EGFP as the template and two pairs of primers: S1, 5′-GTCCTGATACCTCTACAGCTGCTCAGGC-3′, A1, 5′-AGGAGATCCTGCCCCGGCACTTCG-3′; S2, 5′-GGGGCAGGATCTCCTGTCATCTCACC-3′, A2, 5′-TAGAGGTATCAGGACCACTATGAAGTAGAAGG-3′. The PCR products were ligated using the In-Fusion HD Cloning Kit (Takara Bio USA, Inc., San Jose, CA, USA). All cDNA inserts were validated by DNA sequencing.

Cell culture and transfection: HEK293 cells (ATCC) were cultured in Dulbecco’s Modified Eagle Medium (DMEM, high glucose), supplemented with 10% fetal bovine serum (FBS), 100 U/mL penicillin, and 100 µg/mL streptomycin, at 5% CO_2_, 37 °C, in a humidified incubator. They were passaged at 1:5 to 1:10 dilutions after trypsin digestion every 3 to 4 days for routine maintenance. For transfection, the cells were seeded in wells of a 12-well plate and transfected the next day when the cell density reached ~80% confluence. For each well, 1.6 µg of the desired cDNA and 4 µL of Lipofectamine 2000 (Invitrogen, Carlsbad, CA, USA) were separately diluted in 50 µL of Opti-MEM (Invitrogen) for each. After incubation at room temperature (~22 °C) for 5 min, the solutions were combined and thoroughly mixed. The mixture was allowed to settle for 20 min at room temperature before transferring to the cells cultured in the 12-well plate. Then, the plate was returned to the incubator for 6 h before trypsin digestion and reseeding onto polyornithine-coated glass coverslips [41]. The transfected cells were used within 18 to 36 h for experiments. 

Vacuole enlargement and fluorescence imaging: To induce endosome-lysosome fusion for enlarged endolysosome vacuoles used in whole-endolysosome patch-clamp recording, the transfected cells were incubated with 1 µM vacuolin-1 overnight. Differential interference contrast (DIC) and epifluorescence images were taken live with cells placed on the stage of an Olympus BX51WI fluorescence microscope. 

Electrophysiology: Recording pipettes were pulled from borosilicate glass with filament (BF150-86-10, Sutter Instrument, Novato, CA, USA). Voltage clamp recordings were made using an EPC9 or EPC10 amplifier operated with the PatchMaster software (version 1.0.0., both from HEKA Electronik GmbH (Lambrecht (Pfalz), Germany). The electrophysiological data were sampled at 5 kHz and filtered at 2 kHz. All recordings were performed at room temperature.

For inside-out recording, the pipette was filled with the Tyrode’s solution containing (in mM): 145 NaOH, 5 KCl, 1.2 CaCl_2_, 1 MgCl_2_, 10 glucose, 10 HEPES, and 10 MES [2-(N-morpholino)ethanesulfonic acid], with pH adjusted to 7.2, 6.0, or 5.0 using gluconic acid (all solutions had an osmolarity of approx. 290 mOsm). The bath solution had (in mM): 145 KOH, 5 NaCl, 1 MgCl_2_, 0.39 CaCl_2_, 1 EGTA, and 10 HEPES, with pH adjusted to 7.2 with gluconic acid. To obtain sizeable macroscopic currents, low resistance pipettes were used, which had a resistance of ~1 MΩ when filled with the above pipette solution and placed in the bath.

For outside-out recording, the pipette solution contained (in mM) 145 KOH, 5 KCl, 1 MgCl_2_, 1 EGTA, and 10 HEPES, with pH adjusted to 7.4 using gluconic acid. diC8 PI(3,5)P_2_ (1 or 5 µM) was added to the pipette solution immediately before the experiment. The bath was typically Tyrode’s solution. For experiments on ion selectivity, the bath contained (in mM): 150 XCl (X = Na, K or Cs) and 10 HEPES or 105 CaCl_2_ and 10 HEPES (pH 7.2 for all). For pH effects on *Oc*TPC1-R540I and *Oc*TPC3, the pipette solution had (in mM): 145 CsOH, 5 CsCl, 1 EGTA, and 10 HEPES, with pH adjusted to 7.2 with gluconic acid; the bath solutions had (in mM): 145 NaCl, 5 KCl, 2 CaCl_2_, 1 MgCl_2_, 10 HEPES, and 10 MES, with pH adjusted to 7.2, 8.0, 6.0, or 5.0.

For whole-endolysosome recording, the vacuolin-1-treated cells were inspected under an Olympus IX71 epifluorescence inverted microscope to identify single cells with green fluorescence labels on enlarged vacuoles. The cell membrane was cut open with a sharp glass pipette to expose the enlarged vacuole. Then the pipette was switched to a recording pipette filled with the internal solution containing (in mM): 145 NaCl, 5 KCl, 2 CaCl_2_, 1 MgCl_2_, 10 HEPES, and 10 MES, with pH adjusted to 4.6, 6.0, or 7.2 using gluconic acid. The bath solution had (in mM): 145 KOH, 5 NaCl, 1 MgCl_2_, 0.39 CaCl_2_, 1 EGTA, and 10 HEPES, with pH adjusted to 7.2 using gluconic acid. For testing the ion selectivity of *Oc*TPC2, a bath solution with Na-gluconate (Na-Glu) was used, which had (in mM) 145 NaOH, 5 NaCl, and 10 HEPES, with pH adjusted to 7.2 using gluconic acid. The pipette resistance was 8 to 10 MΩ when filled with the above internal solution and placed in the bath. Membrane capacitance of whole endolysosomes was approx. 1 pF. 

For both inside-out and whole-endolysosome recordings, phosphoinositides (all diC8 forms obtained from Cayman Chemical Company, Ann Arbor, MI, USA) were diluted to the final desired concentrations and applied by perfusion towards the recording pipette through a gravity-driven perfusion system. The membrane was held at 0 mV, with voltage ramps from −100 to +100 mV within 200 ms applied every 5 or 10 sec. Step voltage protocols are shown in figure insets and explained in figure legends. 

Data analysis: Concentration–response curves to PI(3,5)P_2_ for *Oc*TPC1 and *Oc*TPC2 were established based on peak current amplitudes at −100 mV obtained from voltage ramps, and that for *Oc*TPC3 was determined from the tail current at 0 mV following a 2-s step pulse to +100 mV. The data points were fit with the Hill Equation: y = A_2_ + (A_1_ − A_2_)/(1 + (x/x_0_)*^n^*), where A_1_ and A_2_ are maximal and minimal responses, respectively, x_0_ is the half-maximal effective concentration (EC_50_), and *n* is Hill coefficient. Conductance–voltage (G–V) relationships were obtained from tail currents at 0 mV following step pulses to different potentials as indicated in figures. The data points were fit with the Boltzmann function: G = G_min_ + (G_max_ − G_min_)/(1 + exp(−(V − V_1/2_)/κ)), where G_max_ and G_min_ are maximal and minimal conductances, respectively, V_1/2_ is the half-maximal activation potential, κ = RT/zf is the slope factor. Inactivation time constants (*τ*) of *Oc*TPC3 tail currents at pH 6.0 and 7.2 were determined by fitting all data points 30 ms after returning to 0 mV with a single exponential function: y = y_0_ + A_1_*exp(−x/*τ*). For current at pH 8.0, the data points were fit with a two-exponential function: y = y_0_ + A_1_*exp(−(x − x_0_)/*τ*_1_) + A_2_*exp(−(x − x_0_)/*τ*_2_). Summary data are expressed as mean ± S.E.M. Statistical analyses were performed using one-sample *t*-test for comparison with a theoretical mean, Student’s *t*-test for two groups, and one-way ANOVA for more than two groups. *p* < 0.05 is considered statistically significant. 

## 3. Results

### 3.1. Rabbit TPC1, TPC2, and TPC3 Are Distantly Related and Exhibit Differential Subcellular Locations When Expressed in HEK293 Cells

Previously, the subcellular localization and function of TPCs were characterized using TPC1 and TPC2 from mice or humans, but TPC3 from other species including rabbit, chicken, clawed frog, and zebrafish [8,11,22,38]. To avoid species-related variations, we thought that it would be better to compare the function and regulation of the three TPC subtypes from a single mammalian species. Previously, we used the cDNA for rabbit (*Oc* for *Oryctolagus cuniculus*) TPC3 [11], but the cDNAs for rabbit TPC1 and TPC2 were unavailable. Therefore, we first cloned them from rabbit kidney total RNA by using RT-PCR. The cloned sequences matched that in the GenBank database by >99.4% for both TPCs at the DNA and protein levels (TPC1, 2440/2454 for DNA, 815/817 for protein, compared to XM_017349553.1; TPC2, 2251/2262 for DNA, 749/753 for protein, compared to GBCT01193833.1). In amino acid sequences, the full-length *Oc*TPC1 is 93.1%, 89.1%, and 88.1% identical to human, mouse, and rat TPC1, respectively, and that of *Oc*TPC2 is 78.5%, 69.8%, and 70.9% identical to human, mouse, and rat TPC2 (Figure 1A). Homology among the three rabbit TPC subtypes is low, showing <30% identity between each pair (TPC1 vs. TPC2, 26.7%; TPC1 vs. TPC3, 27.5%; TPC2 vs. TPC3, 29.9%) in regions that encompass all 12 TM segments and the loop between the two 6-TM repeats. 

To study their function, we subcloned the *Oc*TPC cDNAs into the pEGFP-N1 vector and expressed them in HEK293 cells. For endolysosome patch-clamp recording, the cells were treated with vacuolin-1 to produce enlarged vacuoles. However, as shown in Figure 1B, only *Oc*TPC2 displayed marked labeling on the enlarged vacuoles. Although *Oc*TPC1 and *Oc*TPC3 were also found in intracellular compartments based on the C-terminally tagged GFP, their association with the enlarged vacuoles was much weaker than that of *Oc*TPC2. Instead, they appeared to be associated with the plasma membrane, which was even more pronounced in *Oc*TPC3 than in *Oc*TPC1 (Figure 1B). Thus, for functional analysis, we tried both inside-out patches and whole-endolysosome patches (Figure 2). For inside-out patches, vacuolin-1 was omitted because, in pilot studies, the treatment with vacuolin-1 did not yield obvious differences in basal and PI(3,5)P_2_-evoked currents. 

### 3.2. The Cloned Rabbit TPC1, TPC2, and TPC3 Are Functional Ion Channels Sensitive to PI(3,5)P_2_

In whole-endolysosome and inside-out patches from HEK293 cells that expressed *Oc*TPC1, diC8 PI(3,5)P_2_ (100 nM) evoked similar inwardly rectifying currents with reversal potentials more positive than +70 mV, as revealed by voltage ramps from −100 to +100 mV (Figure 2A). The current amplitudes were comparable between the two methods or slightly larger in inside-out patches. In HEK293 cells that expressed *Oc*TPC2, diC8 PI(3,5)P_2_ (1 µM) also evoked a near-linear current in both whole-endolysosome and inside-out patches when recorded using voltage ramps (Figure 2B), although in inside-out patches, the success rate was low (~30%) and a higher concentration of PI(3,5)P_2_ (10 µM) was needed to elicit sizeable currents. This may be related to the low likelihood of its presence on the plasma membrane as revealed by the GFP fluorescence (Figure 1B). For *Oc*TPC3, the expression in HEK293 cells did not lead to an obvious current when recorded from inside-out patches using voltage ramps from −100 mV to +100 mV either at basal or when diC8 PI(3,5)P_2_ (1 µM) was applied (*data not shown*). Considering that TPC3 from non-mammalian species, zebrafish and clawed frog, acted as voltage-dependent channels that required high positive voltage for activation [22,38], we tried a step pulse protocol, in which a voltage step from 0 mV to +100 mV was applied for 2 s before returning to 0 mV again (Figure 2C, *inset*). In the presence of PI(3,5)P_2_ (1 µM), a huge inward current appeared immediately after returning to 0 mV from the step pulse, which slowly decayed to the baseline; this tail current was undetected or very small in the absence of PI(3,5)P_2_ (Figure 2C).

For concentration dependence on PI(3,5)P_2_, we used inside-out patches for *Oc*TPC1 and *Oc*TPC3, and whole-endolysosome patches for *Oc*TPC2 (Appendix A). All patches showed increases in current amplitudes in response to increasing concentrations of diC8 PI(3,5)P_2_, with *Oc*TPC1 being the most sensitive, which exhibited an obvious response to as low as 3 nM of the phospholipid (Appendix A). By contrast, *Oc*TPC2 and *Oc*TPC3 required ~30 nM diC8 PI(3,5)P_2_ to produce detectable effects (Appendix A). The EC_50_ values were determined to be 26.3 ± 5.0 nM (*n* = 7), 208.5 ± 8.1 nM (*n* = 7), and 142.0 ± 0.2 nM (*n* = 7) for *Oc*TPC1, *Oc*TPC2, and *Oc*TPC3, respectively, based on currents at −100 mV (for *Oc*TPC1 and *Oc*TPC2) and peak tail current at 0 mV following a step pulse to +100 mV (for *Oc*TPC3) (Figure 2D).

Taken together, all three cloned rabbit TPC isoforms form functional, PI(3,5)P_2_-activated, ion channels when heterologously expressed in HEK293 cells, with a rank order of PI(3,5)P_2_ sensitivity of *Oc*TPC1 > *Oc*TPC3 > *Oc*TPC2. However, the differences in lipid compositions between the plasma membrane and endolysosomal membrane and the recording protocols used could also contribute to differences in the measured EC_50_ values. The main differences among them are subcellular localization and voltage dependence.

### 3.3. Rabbit TPC1, TPC2, and TPC3 Display Different Selectivities for Phosphoinositides 

Next, we examined the selectivity of rabbit TPCs among phosphoinositides. We applied the same concentrations (100 nM for *Oc*TPC1, 1 µM for *Oc*TPC2 and *Oc*TPC3) of PI, PI3P, PI4P, PI5P, PI(3,4)P_2_, PI(3,5)P_2_, PI(4,5)P_2_, and PI(3,4,5)P_3_ (all diC8 forms) to the cytoplasmic side of the patch. For *Oc*TPC1 and *Oc*TPC3 we used inside-out patches, while for *Oc*TPC2 we employed whole-endolysosome patches, all prepared from HEK293 cells after transient transfection. Interestingly, while *Oc*TPC2 only responded to PI(3,5)P_2_, *Oc*TPC1 also responded to PI3P (Figure 3A,B), with a mean current amplitude equivalent to 42.1 ± 5.0% (*n* = 6) of that evoked by the same concentration of PI(3,5)P_2_ at −100 mV (Figure 3D). Moreover, for *Oc*TPC3, PI3P was almost equipotent as PI(3,5)P_2_ when applied at 1 µM (Figure 3C), reaching 93.8 ± 5.1% (*n* = 6) of that evoked by PI(3,5)P_2_ based on the tail currents at 0 mV (Figure 3D). In addition, *Oc*TPC3 was also activated by PI(3,4)P_2_, PI(4,5)P_2_, and PI(3,4,5)P_3_, although the responses were much weaker, mounting to 29.9 ± 5.2%, 9.0 ± 2.0% and 7.0 ± 2.0%, respectively, (*n* = 6 for all), of that evoked by the same concentration of PI(3,5)P_2_ (Figure 3D). These results suggest that although PI(3,5)P_2_ is the preferred agonist of all three rabbit TPC isoforms, only TPC2 is exclusively dependent on this phosphoinositide for activation. In addition to PI(3,5)P_2_, TPC1 and TPC3 are also activated by PI3P, a precursor of PI(3,5)P_2_, and TPC3 even exhibits sensitivity to the other two phosphatidylinositol bisphosphates, PI(3,4)P_2_ and PI(4,5)P_2_, as well as PI(3,4,5)P_3_, showing the poorest selectivity for the seven phosphoinositide species among the three TPCs.

It has been reported that PI(4,5)P_2_ acts as an antagonist of TRPML1, another endolysosomal channel activated by PI(3,5)P_2_ [42]. To test if PI(4,5)P_2_ also inhibits *Oc*TPCs, we applied PI(4,5)P_2_ (1 µM) after the currents have been elicited by diC8 PI(3,5)P_2_ (100 nM). For both *Oc*TPC1 and *Oc*TPC2, PI(4,5)P_2_ suppressed the currents evoked by PI(3,5)P_2_ in whole-endolysosome patches by 79.6 ± 4.4% (*n* = 8) and 67.6 ± 5.3% (*n* = 6), respectively (Figure 3E; Appendix A); however, in inside-out patches, the inhibition on *Oc*TPC1 was less pronounced, reaching only 28.0 ± 7.4% (*n* =7) with 1 µM PI(4,5)P_2_ against 100 nM PI(3,5)P_2_ (Figure 3E). Yet, for *Oc*TPC3 in inside-out patches, PI(4,5)P_2_ did not show an inhibitory effect (Figure 3E; Appendix A). These results suggest that PI(4,5)P_2_ is only inhibitory to *Oc*TPC1 and *Oc*TPC2, but not to *Oc*TPC3, and the inhibitory effect is more pronounced in endolysosomal membranes than in the plasma membrane. 

### 3.4. Rabbit TPC1, TPC2, and TPC3 Are Highly Na^+^-Selective When Activated by PI(3,5)P_2_


When stimulated with PI(3,5)P_2_, TPC1 and TPC2 from both humans and mice exhibited a strong Na^+^ selectivity over K^+^ [7,16,29,43]. The Na^+^ selectivity of *Oc*TPC1 and *Oc*TPC2 was also evident based on the high positive reversal potentials recorded in the physiological solutions (Figure 2 and Figure 3), where the cytoplasmic side had mainly K^+^ (145 mM K^+^, 5 mM Na^+^) while the extracellular or endolysosomal luminal side contained mainly Na^+^ (145 mM Na^+^, 5 mM K^+^). To further confirm the Na^+^ selectivity of *Oc*TPC1 and *Oc*TPC2, we performed ion substitution experiments, with the bath solution switched to the one that contained 150 mM Na^+^ after the current was elicited by PI(3,5)P_2_ (1 µM) in the regular cytoplasmic solution. As expected, the switch of the major cation from K^+^ to Na^+^ at the cytoplasmic side dramatically shifted the reversal potentials towards zero mV and increased the outward currents at positive potentials, while the currents at ~100 mV were little affected (Figure 4A,B). These suggest that *Oc*TPC1 and *Oc*TPC2 are Na^+^-selective channels just like their human and mouse counterparts, at least when activated by PI(3,5)P_2_.

For *Oc*TPC3, we first used outside-out patches with 5 µM PI(3,5)P_2_ included in the pipette solution to maintain a high level of channel activation. After the tail current became stable in the normal extracellular solution (Na^+^-Tyrode’s: 145 Na^+^, 5 mM K^+^), the bath was changed sequentially to equiosmotic solutions that contained 150 mM NaCl, 150 mM KCl, 105 mM CaCl_2_, and 150 mM CsCl. The switch to 150 mM NaCl markedly increased the tail current, whereas that to the other equiosmotic solutions abolished the tail current (Figure 4C). For the reversal potential of *Oc*TPC3 in regular cytoplasmic and extracellular solutions, we used inside-out patches with the bath containing 1 µM PI(3,5)P_2_. After the step pulse at +100 mV for 4 s, the voltage was switched to 0, 40, 80, 120, and 160 mV for 100 ms to detect tail currents at different post-step potentials (Figure 4D). Blotting the current–voltage (I–V) relationship of these currents revealed a reversal potential of +85.3 mV (95% confidence: 77.6 to 93.2 mV) (Figure 4E), which matches the theoretical Na^+^ reversal potential (85.6 mV) under these experimental conditions. These results demonstrate that all three rabbit TPCs are highly Na^+^ selective channels when activated by PI(3,5)P_2_. 

### 3.5. OcTPC1 and OcTPC3 Are Voltage-Dependent While OcTPC2 Is Not 

Different TPC subtypes exhibit remarkable differences in voltage dependence [4,8,16,22,29,38]. To characterize the voltage dependence of rabbit TPCs, we employed step pulse protocols to examine the kinetic behaviors of *Oc*TPC-mediated current at different potentials (Figure 5). For *Oc*TPC1 in inside-out patches, the PI(3,5)P_2_-evoked currents displayed obvious inactivation at negative potentials, both the rate and degree of which increased as the voltage changed to more negative such that at −100 mV the current inactivated >70% at the end of the 2-s pulse (Figure 5A). The conductance–voltage (G–V) curve, derived from the tail currents at 0 mV, demonstrated the voltage dependence of *Oc*TPC1, with a V_1/2_ of −63.9 ± 3.8 mV and a slope factor of 34.3 ± 2.8 in the presence of 1 µM PI(3,5)P_2_ (Figure 5F). The large slope factor is comparable to that of some TRP channels but much larger than that of classical voltage-gated channels [44], indicating a weak voltage dependence. Consistent with this interpretation, it has been reported that only the second, but not the first, 6-TM repeat of human and mouse TPC1 confers voltage dependence, through positively charged arginine residues in the S10 (or S4 of the second repeat) TM segment, specifically R539 (for human)/R540 (for mouse) [6,30]. In *Oc*TPC1, the equivalent residue is R540. By changing R540 to isoleucine, the residue in the equivalent position of all mammalian TPC2 [6,7], we completely abolished the inactivation of *Oc*TPC1 at negative potentials (Figure 5B), and the G–V curve also became flat in the voltage range of −180 mV to +60 mV (Figure 5F). These results confirm the voltage sensitivity of *Oc*TPC1 and the critical role of R540. 

For *Oc*TPC2 in whole-endolysosome patches, the application of PI(3,5)P_2_ (100 nM) evoked an inward current at the holding potential of 0 mV. After stepping to other potentials, the current immediately decreased or increased in accord with the direction and degree of the voltage change and remained relatively constant during the voltage pulse (Figure 5C). Similar voltage-independent responses were also obtained in inside-out patches (Figure 5D). The G–V relationship deduced from the whole-endolysosome recording showed a lack of voltage dependence for *Oc*TPC2 between −140 mV and +100 mV (Figure 5F), consistent with previous studies showing the lack of voltage dependence in PI(3,5)P_2_-evoked human and mouse TRPC2 currents [16,29].

For *Oc*TPC3, we applied voltage steps from −30 mV to +90 mV to inside-out patches (Figure 5E, *inset*). The leak currents measured before the application of PI(3,5)P_2_ had a reversal potential of ~0 mV and showed no obvious voltage dependence. After the application of PI(3,5)P_2_ (1 µM), large tail currents developed following the steps to positive potentials (Figure 5E). In addition, because of the Na^+^ selectivity, the TPC3 currents are inward at these positive potentials; thus, their development during the step potentials appeared as a time-dependent decrease in the leak current (Figure 5E). The slow time courses indicate slow kinetics of *Oc*TPC3 activation at the positive potentials, consistent with the previous findings in *Xenopus* TPC3 [38]. Based on the tail currents measured at 0 mV, we constructed the G–V curve for *Oc*TPC3, showing voltage-dependent activation in the presence of 1 µM PI(3,5)P_2_, with a V_1/2_ of 70.1 ± 2.4 mV and a slope factor of 18.9 (Figure 5F), indicating a slightly steeper voltage dependence than *Oc*TPC1.

### 3.6. Rabbit TPCs Have Different pH Dependence

Endolysosomal channels experience different pH environments along the endocytic pathways, with the luminal pH in the lysosome being the most acidic (~pH 5.0) and the extracellular pH being neutral (pH 7.4). It was reported that the endolysosome localized TRPML1 and TRPML3 have different pH sensitivities [45,46] and human TPC1, which is preferentially localized in endosomes that have luminal pH values around 6.0 to 6.5, underwent a dramatic positive shift of its voltage dependence in whole-endolysosome recordings when luminal pH decreased one pH unit from 5.6 to 4.6 [30]. Likewise, in whole-cell recordings, mouse TPC1 also exhibited a positive shift in its voltage dependence and a decrease in its maximal conductance as the extracellular pH changed from neutral to acidic [6]. To test whether this is also the case for *Oc*TPC1, we measured the voltage dependence of *Oc*TPC1 in inside-out patches, with the pipette pH adjusted to 6.0 and 5.0. Similar to the results obtained with the pipette pH being 7.2 (Figure 5A,F), in pH 6.0, *Oc*TPC1 showed a V_1/2_ of −67.3 ± 2.6 mV when the cytoplasmic side was exposed to 1 µM PI(3,5)P_2_ (Figure 6A), suggesting that the activation of *Oc*TPC1 is not affected by acidification at the extracytosolic side until pH 6.0. However, when a pH 5.0 solution was used in the pipette, very small currents were elicited by 1 µM PI(3,5)P_2_ at 0 mV. Stepping to negative potentials, −20 and −40 mV, led to small instantaneous increases in inward currents, which declined gradually (Figure 6B). This confirms that the channel’s open probability was indeed low at 0 mV and it was further decreased at more negative potentials. Stepping to positive potentials also resulted in declining currents between +20 and +80 mV; however, at above +100 mV, the current gradually increased during each of the step pulses (Figure 6B). These were reminiscent of *Oc*TPC3, indicative of activation at high positive voltages. Indeed, the tail currents at 0 mV revealed high open probabilities at potentials >+80 mV, and the G–V curve was shifted dramatically to positive with about 170 mV change of V_1/2_ from −67.3 mV at pH 6.0 to +102.9 ± 4.8 mV at pH 5.0 (Figure 6A). These results indicate that a drop of one pH unit from 6.0 to 5.0 at the extracytosolic side can drastically suppress *Oc*TPC1 activation by PI(3,5)P_2_ within the physiologically relevant membrane potential range.

To evaluate if protons exert their effect on *Oc*TPC1 solely through voltage dependence, we tested how low pH affects the voltage-insensitive mutant *Oc*TPC1-R540I. To record currents from the same patch before and after the pH change, we used outside-out patches, with 1 µM PI(3,5)P_2_ included in the pipette. After stabilization of the currents in the pH 7.2 bath solution, the bath was changed to a solution of pH 6.0 (Figure 6C). At both pH 7.2 and pH 6.0, *Oc*TPC1-R540I exhibited stable currents at all potentials tested (−160 to +60 mV, 1-s steps), with no delay in current activation or obvious inactivation in any of the voltages. Interestingly, although the tail currents remained constant, at negative step potentials, the current increased progressively more as the step voltage became more negative (Figure 6C), producing an inwardly rectifying I–V curve and the rectification was slightly stronger at pH 6.0 than at pH 7.2 (Figure 6C). The inward rectification was also detected using the voltage ramp protocol (Figure 6D). However, the currents at positive potentials remained unchanged by the drop of pH from 7.2 to 6.0. Importantly, when the extracellular pH was changed to pH 5.0, nearly no inward current remained and currents at positive potentials were generally positive (Figure 6C, *right*), suggesting that they may not be Na^+^ selective and therefore not mediated by *Oc*TPC1. Based on currents recorded at −100 mV using the voltage ramps, we determined the activities of *Oc*TPC1-R540I at pH 7.2 and pH 5.0 to be 85 ± 4% and 11 ± 3%, respectively, of that at pH 6.0 (Figure 6E). Thus, in the narrow pH range of 6.0 to 5.0, *Oc*TPC1-R540I switched from being highly active to nearly nonfunctional, suggesting a very tight, voltage-independent, regulation by protons. 

Unlike *Oc*TPC1, *Oc*TPC2 showed a strong preference for low luminal pH, with the current amplitudes being the highest at pH 4.6, which was moderately decreased at pH 6.0, but more dramatically decreased at pH 7.2, as recorded using voltage ramps from enlarged endolysosome vacuoles (Figure 7A,B). This suggests that *Oc*TPC2 is more active in lysosomes, where the luminal pH is lower than in endosomes or on the plasma membrane. 

For pH dependence of *Oc*TPC3, we used outside-out patches, with 5 µM PI(3,5)P_2_ included in the pipette solution. The bath solution was changed from pH 7.2 to pH 6.0 and then to pH 8.0. Interestingly, in the pH 8.0 solution, the tail current at 0 mV showed a larger peak and slower recovery than that at pH 7.2 (Figure 7C–E). In the pH 6.0 solution, the tail current became very small and quickly inactivated (Figure 7C). The peak of the tail current at pH 8.0 and pH 6.0 was 177 ± 12% and 32 ± 6%, respectively, of the peak tail current at pH 7.2 (Figure 7E). Based on the tail currents, we also constructed G–V curves for *Oc*TPC3 at the three pH conditions, showing V_1/2_ values of 53.7 ± 2.8 mV for pH 8.0, 70.1 ± 2.4 mV for pH 7.2, and 105 ± 13 mV for pH 6.0 (Figure 7F). Therefore, *Oc*TPC3 appears to prefer alkaline pH and may be quickly inhibited by the acidic pH in the lumens of both endosomes and lysosomes. Different from *Oc*TPC1, however, *Oc*TPC3 is more sensitive to the pH range of 7.2 to 6.0.

## 4. Discussion

### 4.1. The Three Rabbit TPCs form PI(3.5)P_2_-Activated Na^+^ Channels with Different Selectivity to Phosphoinositide Species

TPCs are unique members of the voltage-gated ion channel superfamily because of their representation of the evolutionary intermediary between the single Shaker-like 6-TM domain K^+^ channels and nonselective cation channels and the four-repeats 24-TM segment voltage-gated Na^+^ and Ca^2+^ channels, as well as their preferential distribution in intracellular organelles. The presence of TPC in all plants also indicates a common ancestor dating back prior to the separation between animals and plants [2,3,9]. Although sharing the same membrane topology, the three mammalian TPCs are quite diverse among themselves in their amino acid sequences and subcellular distributions. While the lysosomal localization of mammalian TPC2 is well established, the exact subcellular locations of TPC1 and TPC3 remain somewhat uncertain although they are likely associated with certain populations of endosomes [11,12,13]. Both endosomal and lysosomal membrane proteins are able to traffic to the plasma membrane during the processes of endosome recycling and lysosome exocytosis. Therefore, it is not surprising that some of the endolysosomal residential proteins are found on the plasma membrane, especially when overexpressed. For TPC2, plasma membrane retention has often been enhanced by mutating the two N-terminal leucine residues, L11/L12, to alanine [37,47], indicating that trafficking to the plasma membrane represents a normal part of the life cycle of this channel. For *Oc*TPC1 and *Oc*TPC3, the overexpression leads to their obvious presence on the plasma membrane and readily detectable PI(3,5)P_2_-evoked currents in inside-out patches. However, although the presence of the heterologously expressed *Oc*TPCs on the plasma membrane provided the convenience for electrophysiological analysis, it does not necessarily suggest that these channels exert their physiological function at the cell surface. 

Despite the sequence diversity and likely functional distinction because of the different subcellular localization, we found that all three *Oc*TPCs are sensitive to PI(3,5)P_2_, a phosphoinositide species mainly found in the membranes of acidic organelles [48]. Among them, *Oc*TPC1 appears to be the most sensitive to PI(3,5)P_2,_ exhibiting an EC_50_ of ~26 nM in inside-out patches. Given that *Oc*TPC1 is inhibited by PI(4,5)P_2_, which is relatively enriched in the plasma membrane, the affinity of *Oc*TPC1 to PI(3,5)P_2_ could be even higher. Still, our measured EC_50_ value is slightly lower than that reported for mouse TPC1 (145 nM for the R540Q mutant) analyzed by whole-cell recordings [6]. Additionally, we found *Oc*TPC1 to be activatable by PI3P, the precursor of PI(3,5)P_2_ that is abundantly present in the membranes of acidic organelles [48]. Even though PI3P is less potent than PI(3,5)P_2_, it could be a more relevant endogenous activator of TPC1 in the endosomal membrane because of its greater abundance. 

On the other hand, *Oc*TPC2 was only activated by PI(3,5)P_2_ but not any of the other phosphoinositide species. This strict selectivity on PI(3,5)P_2_ is consistent with the previous studies on human TPC2 [7,29]. The measured EC_50_ of PI(3,5)P_2_ for *Oc*TPC2, 208.5 nM, is comparable to the reported values for human TPC2, which ranged from 390 to ~1000 nM [7,29,49]. In addition, we found that the PI(3,5)P_2_-evoked *Oc*TPC2 current was inhibited by PI(4,5)P_2_. Such an inhibitory effect may contribute partly to the low success rate of and higher PI(3,5)P_2_ concentration needed for detecting *Oc*TPC2 activation in inside-out patches, in addition to the low abundance of the channel protein on the plasma membrane.

Surprisingly, we found *Oc*TPC3 to be the least selective on phosphoinositide species, with PI3P being almost equipotent as PI(3,5)P_2_. Although the initial work on zebrafish TPC3 showed that the channel was activated by high positive voltages, but not PI(3,5)P_2_ or PI(4,5)P_2_ [22], a latter study using *Xenopus* TPC3 indicated that TPC3 requires PI(3,4)P_2_ or PI(3,5)P_2_ for its activation by voltage [38]. In the current study, *Oc*TPC3 exhibited no voltage-dependent activation in inside-out patches in the absence of any added phosphoinositides, but became activated at high positive voltages in the presence of not only PI(3,5)P_2_ or PI3P, but also PI(3,4)P_2_, PI(4,5)P_2_, or PI(3,4,5)P_3_. This suggests that TPC3, or at least mammalian TPC3, is sensitive to a broad range of phosphoinositide species that have C3 phosphate although the affinities may vary. This broad ligand range implicates that TPC3 may exert function in multiple endolysosomal compartments and plasma membranes. 

Recent cryo-EM structures of mouse TPC1 and human TPC2 revealed that PI(3,5)P_2_ binds to the junction at S3, S4, and S4–S5 linker of the first 6-TM repeat, with additional residues from S6 of the same repeat involved for channel gating [6,7]. While the inositol 1,3,5-trisphosphate head group of the ligand is situated at the cytoplasmic side, the acyl chains are inserted upright into the membrane [6]. All the key residues involved in PI(3,5)P_2_ binding are conserved among humans, mice, and rabbits for TPC1 and TPC2, respectively. However, differences exist between TPC1 and TPC2 in all the regions involved in PI(3,5)P_2_ binding, including the pre-S1, S3, S4, S4–S5 linker, and S6 regions of the first repeat. The key residues involved in PI(3,4)P_2_ regulation of *Xenopus* TPC3 have also been examined through site-directed mutagenesis [38]. These residues are conserved between *Xenopus* and rabbit TPC3. Importantly, except for the conserved RRxxR (x means any amino acid; “R” is arginine in TPC1 and TPC3, but lysine in TPC2) motif in the S4-S5 linker, other key residues tend to differ among TPC1, TPC2, and TPC3. Future studies may explore these differences and determine how they affect the affinity and selectivity of TPCs to various phosphoinositide species. 

Furthermore, in line with the previous studies [16,29,30,32,33,49], the PI(3,5)P_2_-evoked TPC currents is Na^+^ selective. However, this does not exclude that under other conditions; for instance, with the stimulation by NAADP or the synthetic ligand TPC2-A1-N, the channels may also conduct Ca^2+^ or even K^+^ [32,33,34,50]. It would be of interest in future studies to test how the three *Oc*TPCs respond to NAADP in the absence and presence of the newly identified NAADP-binding proteins [35,36,37], how they respond to the recently described TPC synthetic agonists [34,40], and whether and how ion selectivity and other regulatory properties change compared to that activated by the phosphoinositide.

### 4.2. The Three Rabbit TPCs Exhibit Distinct Voltage and pH Dependence

In line with the idea that vertebrate TPCs are co-regulated by phospholipid and voltage [6,39], we found that both *Oc*TPC1 and *Oc*TPC3 are codependent on PI(3,5)P_2_ and depolarizing voltage for activation. In the absence of PI(3,5)P_2_, minimal or no Na^+^-selective current was detected in inside-out patches excised from cells that expressed *Oc*TPC1 or *Oc*TPC3. However, PI(3,5)P_2_ only activated *Oc*TPC1 and *Oc*TPC3 at the permissive voltages, which for *Oc*TPC3 are much higher than for *Oc*TPC1. Notably, the V_1/2_ of *Oc*TPC1 with 1 µM PI(3,5)P_2_, −63.9 mV at pH 7.2, is much lower than that reported for mouse TPC1 (16.2 mV, 2 µM PI(3,5)P_2_, pH 7.4) in whole-cell [6] and human TPC1 (2.6 mV, 1 µM PI(3,5)P_2_, pH 6.6) in whole-endolysosome patches [30]. While differences in endogenous levels of PI(3,5)P_2_ and PI3P, which may impact the effective ligand concentration experienced by the channel, could influence the V_1/2_, for example, at 0.1 µM PI(3,5)P_2_, *Oc*TPC1 had a V_1/2_ of −45.6 ± 3.1 mV and a slope factor of 31.6 ± 2.4 (*n* = 5), the higher sensitivity of *Oc*TPC1 to PI(3,5)P_2_ (*see above*) and subtle amino acid differences between *Oc*TPC1 and human/mouse TPC1 probably underlie the large negative shift of the voltage dependence of *Oc*TPC1. 

More interesting is the very large shift of the voltage dependence to positive potentials when extracellular (equivalent to endosome lumen) pH changed from 6.0 to 5.0. This one-pH unit drop caused a 170-mV positive shift of the V_1/2_ of *Oc*TPC1, which is much greater than the 63-mV depolarization shift reported for human TPC1 when luminal pH changed from 5.6 to 4.6 [30]. However, between pH 7.2 and pH 6.0, the voltage dependence of *Oc*TPC1 did not change significantly, and if anything, the V_1/2_ actually decreased by 3.4 mV. This also differs greatly from human TPC1 studied in the whole-endolysosome configuration, where the V_1/2_ increased by 25.6 mV as the luminal pH decreased from 6.6 to 5.6 [30] and mouse TPC1 studied in whole-cell configuration, where the V_1/2_ increased by 22 mV between pH 7.4 and 6.0 [6]. Therefore, unlike human TPC1, the rabbit TPC1 has a very steep pH dependence between pH 6.0 and 5.0, suggesting a dramatic decrease in activity as the channel traffics from endosomes to the lysosome through the endocytic process. Even taking into account that pH 5.0 is slightly outside the buffering range of MES and therefore the solution pH could have an error, the resulting pH would unlikely fall below 4.5 and thus be well within the pH range of the lysosome lumen. Additionally, it is important to note that pH 5.0 suppresses *Oc*TPC1 function not only by shifting the voltage dependence dramatically to high positive potentials but also by greatly reducing the maximal response. With the voltage-insensitive TPC1 mutant, a drop of the extracytosolic pH from 6.0 to 5.0 decreased the PI(3,5)P_2_-evoked current by nearly 90%. 

Consistent with previous studies [7,16,29,30], the voltage dependence is lost in *Oc*TPC2. Cryo-EM studies have revealed that the voltage dependence of TPCs arises from an array of arginines lining at one side of the S4 TM segment of the second 6-TM repeat in a 3_10_ helix [4,6,7,8]. In canonical voltage-gated channels, there can be up to five S4 gating charge residues, R1–R5. However, in TPC1, only R3 and R5 are arginines and both of them are critical for voltage gating, through a sliding movement of the S4 that places either R3 or R5 arginine in the gating-charge transfer center formed by residues in the S2 and S3 segments of the second 6-TM repeat at either resting or activated state, respectively [6]. Thus, the mutation of the R3 arginine, R540, to a neutral residue forces the R5 arginine to be permanently associated with the gating-charge transfer center, allowing full channel activation even at hyperpolarized potentials [6,30]. This explains the voltage independence of the *Oc*TPC1-R540I mutation. In essence, R540I of TPC1 mimics the R3 isoleucine of TPC2. Therefore, although TPC2 has R4 and R5 arginines, the lack of R3 arginine still places R5 arginine in the gating-charge transfer center to keep the second 6-TM repeat in the activated state [7], making TPC2 voltage-independent, although the actual activation still awaits PI(3,5)P_2_ binding in the first 6-TM repeat. 

Previously, it was reported that increasing the luminal pH from 4.6 to 7.4 had minimal effects on PI(3,5)P_2_-evoked activation of human TPC2 [29]. However, we found *Oc*TPC2 to be inhibited by ~92% when luminal pH increased from 4.6 to 7.2 or ~90% when luminal pH increased from 6.0 to 7.2. Between pH 4.6 and pH 6.0, the current did not change significantly, suggesting that the most dramatic channel inhibition occurs between pH 6.0 and 7.2. Given that pH 4.6 and 6.0 represent the luminal acidity of lysosomes and late endosomes, respectively, this finding suggests that *Oc*TPC2 is the most active in these acidic organelles, which is also consistent with the subcellular localization of TPC2 [1,11]. However, the pH dependence of *Oc*TPC2 is in sharp contrast with that of *Oc*TPC1-R540I, despite the voltage independence of both, further stressing that the pH dependence is a feature determined by other regions of the TPC proteins unrelated to the voltage gating. The mechanisms of pH regulation of TPCs warrant further investigation.

The high voltage dependence of nonmammalian TPC3 has been studied previously [8,22,38]. Here, we confirm that *Oc*TPC3 is also high voltage-dependent, with a V_1/2_ of 70.1 mV and a slope factor of 18.9 at pH 7.2 in the presence of 1 µM PI(3,5)_2_ in inside-out patches. These values are not very different from that reported for zebrafish TPC3 (V_1/2_ = 76.9 mV, κ = 18.2, Ref. [22]) and *Xenopus* TPC3 (V_1/2_ = 62–64 mV, κ = 21–26, Refs. [38,39]). Like zebrafish TPC3, *Oc*TPC3 contains arginines in all three positions, R3, R4, and R5, in the S4 segment of the second 6-TM repeat. Previously, all three arginines were reported to be involved in the voltage-gating of TPC3, and in the closed state, the R4 arginine is associated with the tyrosine in the gating-charge transfer center [8]. The slightly steeper slope of TPC3 than TPC1 is consistent with the larger gating charge. Whether this R4 arginine-tyrosine interaction stabilizes TPC3 in the closed state at resting potentials in the presence of a phosphoinositide ligand and thereby underpins the high voltage requirement of TPC3 warrants additional investigation. 

Finally, in line with the report that low extracellular pH (pH 4.6) suppressed *Xenopus* TPC3 [38], we showed that *Oc*TPC3 preferred more alkaline extracytosolic pH. Even with the acidification from pH 7.2 to pH 6.0, the activity decreased by ~68%, along with a positive V_1/2_ shift of ~35 mV, but with the alkalinization to pH 8.0, the activity increased by ~77%, along with a negative V_1/2_ shift of ~16 mV. Most likely, the low pH inhibited TPC3 by accelerating channel inactivation. Notably, TPC3 is inhibited by the relatively moderate pH drop to the level close to that of endosomes, indicating that this channel tends to lose its activity rather quickly along the endocytic pathway. 

In summary, the three rabbit TPC isoforms share similarities in responding to PI(3,5)P_2_ as an endogenous lipid ligand to conduct Na^+^. However, they exhibit differences in selectivity to other phosphoinositides, such as PI3P, PI(3,4)P_2_, and PI(3,4,5)P_3_. In addition, whereas PI(4,5)P_2_ inhibits TPC1 and TPC2, it weakly activates TPC3. The three TPCs also display different voltage dependence, with TPC1 and TPC3 being codependent on both phosphoinositide and depolarizing voltage and TPC2 being dependent only on PI(3,5)P_2_, but not voltage. Compared to TPC1, TPC3 exhibits a large shift of the voltage dependence to positive potentials, mounting to a 134-mV increase in V_1/2_. Moreover, all three TPCs are sensitive to pH on the extracytosolic side, with TPC2, TPC1, and TPC3 preferring acidic (pH 5–6), acidic to neutral (pH 5–6), and neutral to alkaline (pH 7–8) pH, respectively. The pH sensitivity may be optimally tuned for the channels to exert their function in specific organelles where they reside.

## Figures and Tables

**Figure 1 cells-11-02006-f001:**
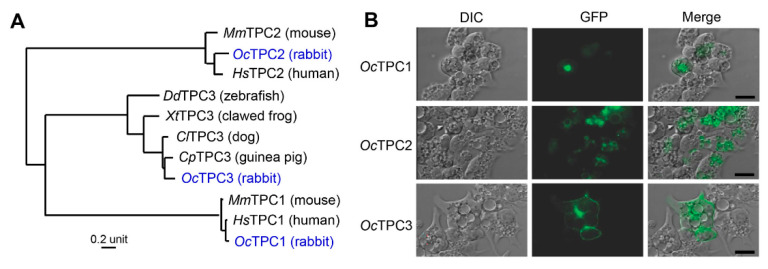
Rabbit TPC1, TPC2, and TPC3 are distantly related and exhibit differential subcellular locations when expressed in HEK293 cells. (**A**) Phylogenetic tree of three vertebrate TPC subtypes, with the rabbit sequences highlighted in blue. Multiple sequence alignment was made using ClustalW (https://www.genome.jp/tools-bin/clustalw (accessed on 11 May 2022)) and the tree generated using the PhyML algorithm. GenBank accession numbers are: TPC1, ON502176 (rabbit), NP_060371.2 (human), NP_665852.1 (mouse); TPC2, ON502177 (rabbit), NP_620714 (human), NP_666318 (mouse); TPC3, EU344155.1 (rabbit); NP_001184074 (dog); XP_013002737 (guinea pig); XM_002940341 (Western clawed frog), NP_001170916 (zebrafish). (**B**) DIC and epifluorescence images of HEK293 cells transiently transfected with cDNA coding for *Oc*TPC1-EGFP, *Oc*TPC2-EGFP, or *Oc*TPC3-EGFP as indicated. Cells were treated with vacuolin-1 (1 µM) overnight before images were taken live without fixation. Scale bars, 20 µm.

**Figure 2 cells-11-02006-f002:**
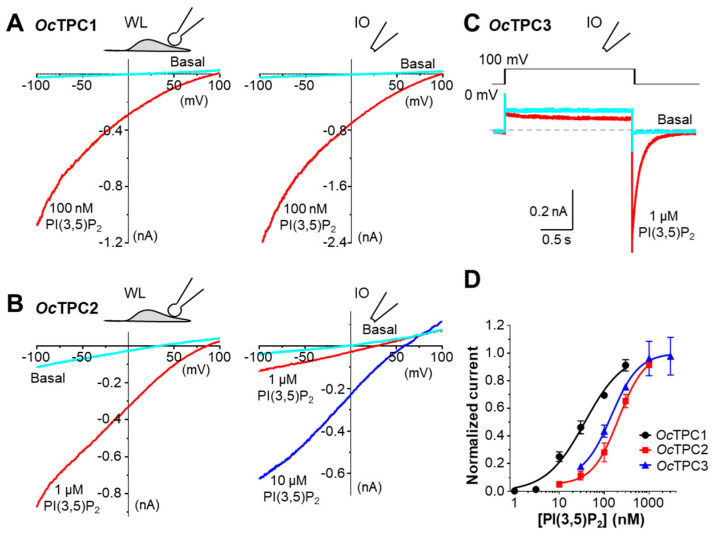
**Rabbit TPC1, TPC2, and TPC3 are activated by PI(3,5)P_2_**. HEK293 cells transiently transfected with cDNA coding for *Oc*TPC1-EGFP, *Oc*TPC2-EGFP, or *Oc*TPC3-EGFP as indicated were used for voltage-clamp recording using either whole-endolysosome (WL) or inside-out (IO) configurations. (**A**,**B**) Representative current–voltage (I–V) curves obtained from voltage ramps for WL (*left*) and IO (*right*) patches from cells that expressed *Oc*TPC1 (**A**) or *Oc*TPC2 (**B**). The patches were untreated (Basal) or exposed to PI(3,5)P_2_ at the indicated concentrations in the bath. (**C**) Representative current traces detected in an IO patch excised from a cell that expressed *Oc*TPC3. The voltage was stepped from the holding potential at 0 mV to +100 mV for 2 s before returning to 0 mV. Dashed line indicates zero current. Note the large tail current in the presence of PI(3,5)P_2_ (*red trace*) in the bath. The square response to +100 mV step under basal conditions (*cyan trace*) represents leak, unrelated to *Oc*TPC3 function. (**D**) Concentration–response curves to PI(3,5)P_2_ for *Oc*TPC1, *Oc*TPC2, and *Oc*TPC3, determined using IO (for *Oc*TPC1 and *Oc*TPC3) and WL (for *Oc*TPC2) patches. Data are means ± SEM for *n* = 7 patches for each. Solid lines represent fit to the Hill equation.

**Figure 3 cells-11-02006-f003:**
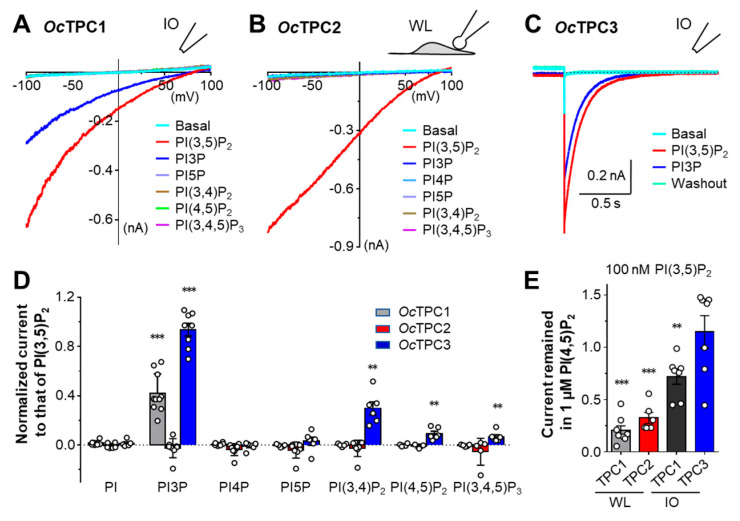
**Rabbit TPC1, TPC2, and TPC3 display different selectivities for phosphoinositides.** HEK293 cells transiently transfected with cDNA coding for *Oc*TPC1-EGFP, *Oc*TPC2-EGFP, or *Oc*TPC3-EGFP as indicated were used for voltage-clamp recording using either inside-out (IO, for TPC1 and TPC3) or whole-endolysosome (WL, for TPC2) configurations. (**A**) Representative I–V curves derived from voltage ramps for *Oc*TPC1 in an IO patch exposed to 100 nM indicated phosphoinositides in the bath. (**B**) Representative I–V curves derived from voltage ramps for *Oc*TPC2 in a WL patch exposed to 1 µM indicated phosphoinositides in the bath. (**C**) Representative current traces of *Oc*TPC3 detected in an IO patch exposed to 1 µM indicated phosphoinositides in the bath. The voltage was stepped from the holding potential at 0 mV to +100 mV for 4 s before returning to 0 mV. Only the last 0.3 s of the +100-mV step and the tail at 0 mV are shown for clarity. Dashed line indicates zero current. (**D**) Summary of peak currents at −100 mV (for TPC1 and TPC2) and tail currents at 0 mV (for TPC3) elicited by the indicated phosphoinositides normalized to that by PI(3,5)P_2_. Data are means ± SEM for *n* = 3–11 patches. White circles represent individual data points. ** *p* < 0.01, *** *p* < 0.001, different from the theoretical mean of 0 by one-sample *t*-test. (**E**) Current in the presence of 100 nM PI(3,5)P_2_ plus 1 µM PI(4,5)P_2_ normalized to that in 100 nM PI(3,5)P_2_ for WL and IO patches obtained from cells that expressed *Oc*TPC1, *Oc*TPC2, or *Oc*TPC3 as indicated. For TPC1 and TPC2, currents at −100 mV obtained from voltage ramps were analyzed; for TPC3, tail currents from 0 mV following the step to +100 mV were analyzed. Data are means ± SEM for *n* = 6–8 patches. ** *p* < 0.01, *** *p* < 0.001, different from the theoretical mean of 1 by one-sample *t*-test.

**Figure 4 cells-11-02006-f004:**
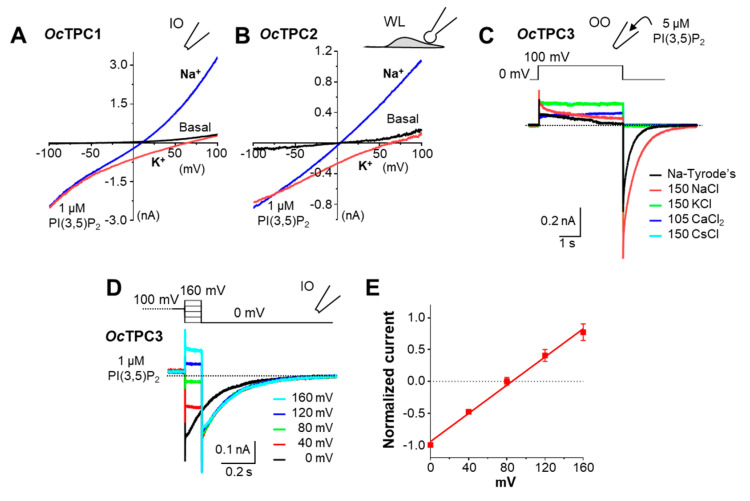
**All three TPC subtypes from rabbit are Na^+^ selective when activated by PI(3,5)P_2_.** HEK293 cells were transiently transfected with cDNA coding for *Oc*TPC1-EGFP, *Oc*TPC2-EGFP, or *Oc*TPC3-EGFP. (**A**,**B**) Representative I–V curves derived from voltage ramps for *Oc*TPC1 in an inside-out (IO) patch (**A**) and *Oc*TPC2 in a whole-endolysosome (WL) patch (**B**) sequentially exposed to 1 µM PI(3,5)P_2_ in the normal bath solution containing mainly K^+^ (*red traces*) and then in a Na^+^-based bath solution (*blue traces*). (**C**) Representative current traces of *Oc*TPC3 detected in an outside-out (OO) patch containing 5 µM PI(3,5)P_2_ in the pipette solution and sequentially exposed to normal Tyrode’s solution followed by equiosmotic solutions that contained 150 mM NaCl, 150 mM KCl, 105 mM CaCl_2_, and 150 mM CsCl. Note the lake of tail currents in KCl, CaCl_2_, and CsCl. (**D**) Representative current traces of *Oc*TPC3 detected in an IO patch exposed to 1 µM PI(3,5)P_2_ in normal Tyrode’s solution. After stepping to +100 mV for 4 s, the voltage was switched to 0, 40, 80, 120, and 160 mV for 100 ms before returning to 0 mV. For clarity, only the last 0.1 s of the +100-mV step is shown together with the tail currents at post-step potentials of 0 to 160 mV followed by that at 0 mV. For (**C**,**D**), dashed lines indicate zero current. (**E**) I–V relationship of currents at the post-step potentials normalized to that of the absolute value at 0 mV for patches recorded as in (**D**). Data points are means ± SEM of *n* = 5 patches fit by linear regression.

**Figure 5 cells-11-02006-f005:**
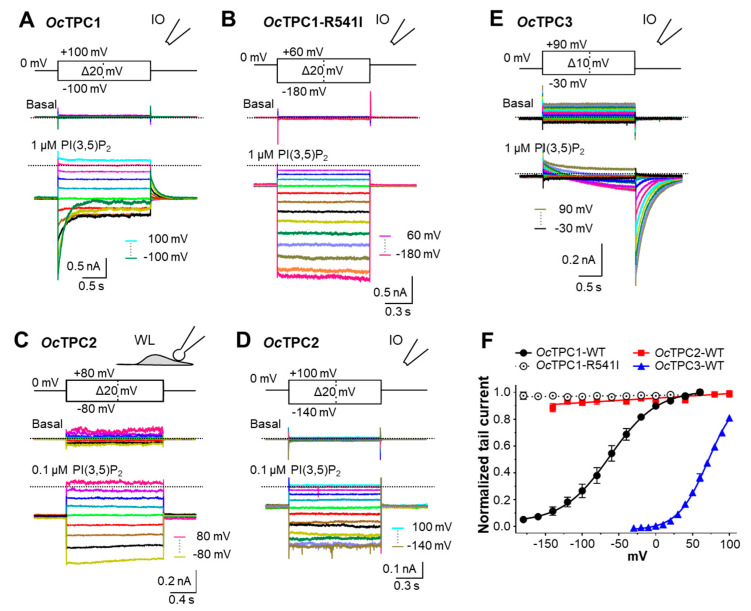
**The three rabbit TPC isoforms display different voltage dependence.** HEK293 cells were transiently transfected with cDNA coding for *Oc*TPC1-EGFP, *Oc*TPC1-R540I-EGFP, *Oc*TPC2-EGFP, or *Oc*TPC3-EGFP. (**A**,**B**) Representative currents elicited by voltage steps from −100 mV to +100 mV with 20-mV increments (**A**, see inset) and from −180 mV to +60 mV with 20-mV increments (**B**, see inset) in inside-out (IO) patches excised from cells that expressed wild type *Oc*TPC1 (**A**) and its mutant, *Oc*TPC1-R540I (**B**). Note the fast inactivation at negative potentials for wild-type *Oc*TPC1 (**A**) but no inactivation for the *Oc*TPC1-R540I mutant even at −180 mV. (**C**,**D**) Representative currents elicited by voltage steps from −80 mV to +80 mV with 20-mV increments (**C**, see inset) and from −140 mV to +100 mV with 20-mV increments (**D**, see inset) in whole-endolysosome (**C**) and IO (**D**) patches from cells that expressed *Oc*TPC2. (**E**) Representative currents elicited by voltage steps from −90 mV to +30 mV with 10-mV increments (see inset) in an IO patch excised from a cell that expressed *Oc*TPC3. Note the leak currents at basal and thus the apparent current decreases in PI(3,5)P_2_ at positive potentials during the step represent increases in inward Na^+^ currents. For (**A**–**E**), holding potential was 0 mV and the patches were either unstimulated (Basal) or exposed to PI(3,5)P_2_ at the indicated concentrations. Horizontal dashed lines represent zero current. (**F**) Conductance–voltage (G–V) curves determined using tail currents as measured in (**A**–**E**). Data points are means ± SEM of *n* = 4 (*Oc*TPC1 and *Oc*TPC1-R540I) and *n* = 6 (*Oc*TPC2 and *Oc*TPC3) patches fit by linear regression (*Oc*TPC1-R540I and *Oc*TPC2) and Boltzmann function (*Oc*TPC1 and *Oc*TPC3). WT, wild type.

**Figure 6 cells-11-02006-f006:**
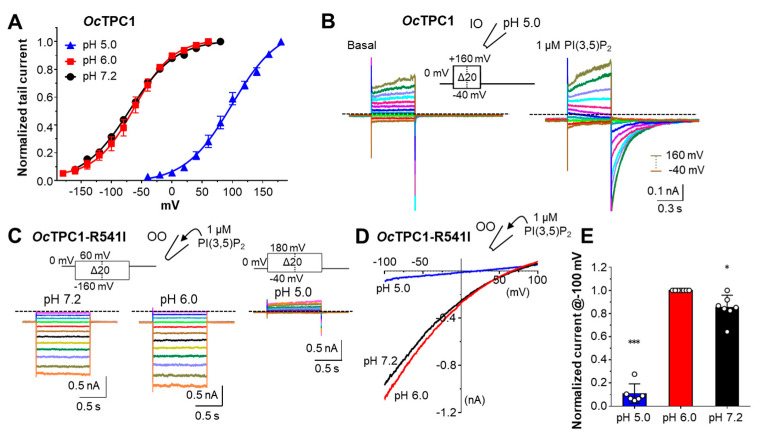
**Low pH inhibits rabbit TPC1 in both voltage-dependent and -independent manners.** HEK293 cells were transiently transfected with cDNA coding for *Oc*TPC1-EGFP or *Oc*TPC1-R540I-EGFP. (**A**) G–V curves of *Oc*TPC1 in inside-out (IO) patches with pipette solutions having pH 7.2, pH 6.0, or pH 5.0 and exposed to 1 µM PI(3,5)P_2_ in the bath. Data points are means ± SEM of *n* = 4 (pH 7.2, same data from Figure 5F) and *n* = 5 (pH 6.0 and pH 5.0) patches fit by Boltzmann function. (**B**) Representative currents elicited by voltage steps from −40 mV to +160 mV with 20-mV increments (see inset) in an IO patch excised from a cell that expressed wild-type *Oc*TPC1. The pipette solution had pH 5.0 and the patch was unstimulated (*left*) or exposed to 1 µM PI(3,5)P_2_ (*right*). (**C**) Representative currents elicited by voltage steps from −160 mV to +60 mV with 20-mV increments (*left* and *middle*, see inset) and from −40 mV to +180 mV with 20-mV increments (*right*, see inset) in an outside-out (OO) patch excised from a cell that expressed *Oc*TPC1-R540I. The pipette solution contained 1 µM PI(3,5)P_2_ and the patch was exposed sequentially to pH 7.2 (*left*), pH 6.0 (*middle*), and pH 5.0 (*right*) bath solutions. For both (**B**,**C**), holding potential was 0 mV and horizontal dashed lines indicate zero current. (**D**) Representative I–V curves derived from voltage ramps for *Oc*TPC1-R540I in an OO patch with the pipette solution containing 1 µM PI(3,5)P_2_ and exposed sequentially to pH 7.2 (*black*), pH 6.0 (*red*), and pH 5.0 (*blue*) bath solutions. (**E**) Summary of currents at −100 mV recorded as in (**D**), normalized to that at pH 6.0. Data are means ± SEM for *n* = 6–7 patches. White circles represent individual data points. * *p* < 0.05, *** *p* < 0.001, different from the theoretical mean of 1 by one-sample *t*-test.

**Figure 7 cells-11-02006-f007:**
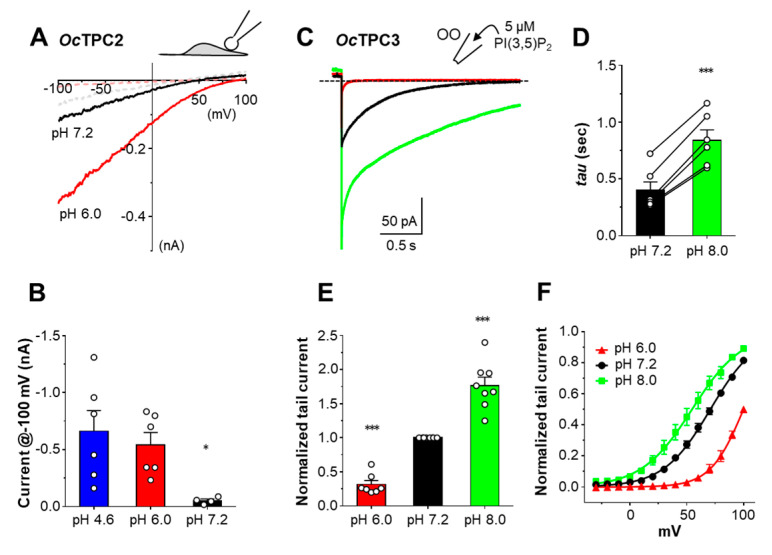
***Oc*TPC2 prefers acidic pH, while *Oc*TPC3 is more active at alkaline extracytosolic pH.** HEK293 cells were transiently transfected with cDNA coding for *Oc*TPC2-EGFP or *Oc*TPC3-EGFP. (**A**) Representative I–V curves derived from voltage ramps for *Oc*TPC2 in whole-endolysosome (WL) patches with pipette solutions having pH 7.2 and pH 6.0. The patch was unstimulated (*dashed lines*) or exposed to 1 µM PI(3,5)P_2_ (*solid lines*). Traces from two different patches were overlaid for comparison between pH 6.0 and pH 7.2. See Figure 2B *left* for an example of pH 4.6. (**B**) Summary of current amplitude at −100 mV recorded as in (**A**). Data are means ± SEM for *n* = 4–6 patches, with individual data points shown as while circles. * *p* < 0.05, different from pH 4.6 by one-way ANOVA with Tukey’s multiple comparisons test. (**C**) Representative current traces of *Oc*TPC3 detected in an outside-out (OO) patch with the pipette solution containing 5 µM PI(3,5)P_2_ and the patch sequentially exposed to pH 7.2 (*black*), pH 6.0 (*red*), and then pH 8.0 (*green*) bath solutions. The voltage was stepped from the holding potential at 0 mV to +100 mV for 4 s before returning to 0 mV. For clarity, only the last 0.14 s of the +100-mV step is shown together with the tail currents at 0 mV. Dashed line indicates zero current. (**D**) Summary of inactivation time constant (*tau*) for tail currents at pH 7.2 and pH 8.0 for *Oc*TPC3 recorded as in (**C**). Data are means ± SEM for *n* = 6 patches. *** *p* < 0.001 by paired *t*-test. (**E**) Summary of tail current normalized to that at pH 7.2 for *Oc*TPC3 recorded as in (**C**). Data are means ± SEM for *n* = 7–8 patches. *** *p* < 0.001, different from the theoretical mean of 1 by one-sample *t*-test. (**F**) G–V curves for patches recorded as in (**C**) but with tail currents obtained after step pulses to different potentials. Data are means ± SEM for *n* = 6 (pH 7.2, same data from Figure 5F), *n* = 4 (pH 8.0), and *n* = 3 (pH 6.0) patches fit by the Boltzmann function.

## Data Availability

Data supporting the reported results are available from the corresponding author upon request. Sequences of *Oc*TPC1 and *Oc*TPC2 are deposited in GenBank with accession numbers ON502176 and ON502177, respectively.

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
