# Peer review of "The Three Two-Pore Channel Subtypes from Rabbit Exhibit Distinct Sensitivity to Phosphoinositides, Voltage, and Extracytosolic pH"

_cells, 2022, doi:10.3390/cells11132006_

Round 1

Reviewer 1 Report

This manuscript from the Zhu lab compares the sensitivity of rabbit TPCs to PIs, voltage and luminal pH. The authors report differential regulation. Overall, the manuscript is well written and the experiments carefully done (vacuolar patch clamp is not easy). But novelty is lacking since nearly all the conclusions have been reached previously. This limits the scope somewhat.

1. A previous study from the Patel lab (PMID: 19940116) comparing the NAADP sensitivity of TPC1-3 (from sea urchin) should be cited.

2. Why did the authors not compare NAADP activation? This would been much more interesting given controversies surrounding its action.

3. Cite PMID: 25406377 and PMID: 20463046 in the context of TPCs as evolutionary intermediates and lineage-specific gene loss.

4. Arabidopsis TPC1 was solved by X-ray crystallography not cryo-EM.

5. Cite appropriate evidence supporting the statement that plant TPC1 is not an orthologue of animal TPCs.

6. For inside-out recordings, HEPES was used to control the pH in the pipette solution down to pH 5. It is unlikely that that HEPES would effectively buffer at acidic pH which could account for some of the steep apparent pH dependencies. This needs stating.

7. Fig. 1A. There are few details concerning how the tree was constructed and its statistical strength.

8. Fig. 1B. The epifluorescence images are too blurry to make any firm conclusions.

9. Fig. 2B. Why does TPC2 support any currents at the PM given its intracellular location?

10. Fig. 4. Did the authors test the Na/Ca ion selectivity of TPC1/2?

11. Discuss discrepancies with the Ren lab concerning PI sensitivity of TPC1

12. Cite Ren lab for inhibition of zebrafish TPC2 by PI(4,5)P2

Author Response

Reviewer 1 Comments and Suggestions for Authors

This manuscript from the Zhu lab compares the sensitivity of rabbit TPCs to PIs, voltage and luminal pH. The authors report differential regulation. Overall, the manuscript is well written and the experiments carefully done (vacuolar patch clamp is not easy). But novelty is lacking since nearly all the conclusions have been reached previously. This limits the scope somewhat.

The novelty of this work lies in the sensitivity of OcTPC1 and OcTPC3 to PI3P and that of OcTPC3 in several other phosphoinositides, as well as the huge differences in pH sensitivities among the three TPCs. These have not been reported before.

  1. A previous study from the Patel lab (PMID: 19940116) comparing the NAADP sensitivity of TPC1-3 (from sea urchin) should be cited.

Thanks for bringing this up. We have cited this work.

  1. Why did the authors not compare NAADP activation? This would been much more interesting given controversies surrounding its action.

Unlike PI(3,5)P2, which increased currents in all patches that contain TPC, NAADP has not been a reliable agonist for activating TPCs in electrophysiological experiments, especially in whole-endolysosome patches. Both the success rate and the elicited current amplitude are quite low, although we occasionally do detect large NAADP-evoked hTPC2 currents in some patches. We are trying to define the conditions that promote NAADP activation of the ionic conductance through TPC2. Once the conditions are defined, it will allow meaning comparison among the three TPC subtypes.

  1. Cite PMID: 25406377 and PMID: 20463046 in the context of TPCs as evolutionary intermediates and lineage-specific gene loss.

Cited

  1. Arabidopsis TPC1 was solved by X-ray crystallography not cryo-EM.

Corrected.

  1. Cite appropriate evidence supporting the statement that plant TPC1 is not an orthologue of animal TPCs.

We mean to say that plant TPC1 is not an orthologue of animal TPCN1 gene, but rather a plant gene that likely shares a common ancestor with all three animal TPCN genes. This point was clearly made in previous publications [Calcraft et al., 2009; Zhu et al., 2010], which are now cited.

  1. For inside-out recordings, HEPES was used to control the pH in the pipette solution down to pH 5. It is unlikely that that HEPES would effectively buffer at acidic pH which could account for some of the steep apparent pH dependencies. This needs stating.

Thanks for bringing up this point. We included both 10 mM HEPES and 10 mM MES to buffer pH for all experiments that involved testing pH dependence. I have corrected the solutions used for inside-out recordings. Sorry for the oversight.

  1. Fig. 1A. There are few details concerning how the tree was constructed and its statistical strength.

We used ClustalW at the indicated website (https://www.genome.jp/tools-bin/clustalw) with default setting. The Tree was made using the PhyML algorithm. I am not sure what other details are needed to explain the method. The purpose of Fig. 1A is to show that the cloned OcTPC1-3 are orthologs of the respective TPCs in other vertebrates.

  1. Fig. 1B. The epifluorescence images are too blurry to make any firm conclusions.

Detailed characterization of mammalian TPCs including rabbit TPC3 by confocal microscopy has been made before [Ogunbayo et al., 2015]. We do not intent to repeat the previous work. The purpose of Fig. 1B is to show where the expressed rabbit TPCs were localized after vacuolin-1 treatment. In order to represent the cells used for electrophysiological experiments, the images were taken from the same rig where the electrophysiological recordings were made. Since the journal only gave us 5 days to submit a revision, we also do not have enough time to produce a new set of images for detailed characterization of subcellular localization of rabbit TPC1 and TPC2.

  1. Fig. 2B. Why does TPC2 support any currents at the PM given its intracellular location?

TPC2 traffics to the PM as a part of their life cycle, although the majority of the protein is present intracellularly. This is the basis for the PM localization of the dileucine mutant of TPC2 [Brailoiu et al., 2010; JBC, 285, 38511-38516], which traffics normally to the PM but exhibits attenuated recycling from the PM to intracellular compartments. Thus, WT TPC2 should be present on the PM although less abundant than the dileucine mutant.

  1. Fig. 4. Did the authors test the Na/Ca ion selectivity of TPC1/2?

No, we did not test this. We do not anticipate the results to be any different from the previous work done with human and mouse TPC1/2.

  1. Discuss discrepancies with the Ren lab concerning PI sensitivity of TPC1

The Ren lab reported that human TPC1 was sensitive to PI(3,5)P2, but not to PI(3,4)P2 and PI(4,5)P2 (Cang et al, Nature Chemical Biology, 2014). Our result on OcTPC1 is consistent with their finding since neither PI(3,4)P2 nor PI(4,5)P2 activated OcTPC1. In this respect, there is no discrepancy between the two studies. Since Cang et al. (2014) did not test PI3P, we cannot call it a discrepancy just because we found that PI3P activated OcTPC1.

  1. Cite Ren lab for inhibition of zebrafish TPC2 by PI(4,5)P2

The Ren lab reported that PI(4,5)P2, as well as PI(3,5)P2, had no effect on zebrafish TPC3, which we have cited in our paper. However, we do not find results from the Ren lab showing that PI(4,5)P2 inhibits zebrafish TPC2.

Reviewer 2 Report

This is a sound and comprehensive analysis of rabbit two pore channels aiming to elucidate the particular role of different channel forms. The study is well designed, the methods appropriate and the paper well structured and written. I have several concerns and suggestions:

1. In terms of methodology, I am interested if the authors considered whether the EGFP fusion to channel proteins may affect their localisation and activity? Has that been tested or have they tried to compare the channels tagged at the other end of the channel?

2. With different properties of different channels in different species, it would be very useful to the reader if the authors would add a table summarising different properties of two pore channels across species and in this study.

3. Minor concern, but the manuscript is at times difficult to read due to the abundant use of abbreviations. 

Author Response

Reviewer 2 Comments and Suggestions for Authors

This is a sound and comprehensive analysis of rabbit two pore channels aiming to elucidate the particular role of different channel forms. The study is well designed, the methods appropriate and the paper well structured and written. I have several concerns and suggestions:

Thank you for supporting our work.

  1. In terms of methodology, I am interested if the authors considered whether the EGFP fusion to channel proteins may affect their localisation and activity? Has that been tested or have they tried to compare the channels tagged at the other end of the channel?

This is an important question. The tests had been done for TPC1 and TPC2 from other mammalian species. We have tested human TPC2 with GFP or mCherry tag at either N- or C-terminus. Neither the localization nor the activity was obviously changed. There are now also ample studies from multiple labs showing that the tagged TPC1 and TPC2 have similar subcellular localization and function as untagged or endogenous TPC1 and TPC2, respectively [see e.g., Wang et al., 2012; Cang et al., 2013; Ruas et al., 2015]. However, for human TPC1, the addition of an epitope tag at the N-terminus resulted in loss of protein expression (our unpublished observation). Therefore, for the current study, we only used C-terminal tagged constructs.

  1. With different properties of different channels in different species, it would be very useful to the reader if the authors would add a table summarising different properties of two pore channels across species and in this study.

Thanks for the suggestion. This would be appropriate for a review article, but for a research article, it would appear to be rather excessive.

  1. Minor concern, but the manuscript is at times difficult to read due to the abundant use of abbreviations. 

All the abbreviations used in this ms are standard and frequently used in the literature. We tried our best not to use new and uncommon abbreviations. I am not sure if spelling out these abbreviations will make the ms easier to read.

Reviewer 3 Report

The manuscript by Feng et al describes a comparative study on the function and regulation of three mammalian TPCs all from rabbit. After expression in HEK293 cells they used enlarged vacuoles, inside-out and outside-out patches to characterize the three channels electrophysiologically. The authors used this platform to study the effect of various phosphatidylinositol-phosphates, membrane potential and pH on channel characteristics. The experiments are well designed and performed thoroughly. In addition the paper is well written and the results are presented in a very comprehensive way. 

TPCs have been characterized thoroughly before but it is important to compare the functionalities of TPCs from the same species to avoid possible interspecies artifacts.

Based on this and the thoroughly performed experiments I suggest that the paper is published in cells. I only have very minor suggestions for improvements.

Minor issues

The authors should include size bars on the bioimages included in the paper.

Do the authors have any data showing that the GFP tag does not influence the measured activities? 

Author Response

Reviewer 3 Comments and Suggestions for Authors

The manuscript by Feng et al describes a comparative study on the function and regulation of three mammalian TPCs all from rabbit. After expression in HEK293 cells they used enlarged vacuoles, inside-out and outside-out patches to characterize the three channels electrophysiologically. The authors used this platform to study the effect of various phosphatidylinositol-phosphates, membrane potential and pH on channel characteristics. The experiments are well designed and performed thoroughly. In addition the paper is well written and the results are presented in a very comprehensive way. 

TPCs have been characterized thoroughly before but it is important to compare the functionalities of TPCs from the same species to avoid possible interspecies artifacts.

Based on this and the thoroughly performed experiments I suggest that the paper is published in cells. I only have very minor suggestions for improvements.

Thank you for the encouragement.

Minor issues

The authors should include size bars on the bioimages included in the paper.

Thanks for bringing up this important point. The scale has now been included in Fig. 1B.

Do the authors have any data showing that the GFP tag does not influence the measured activities? 

This is an important question. The tests had been done for TPC1 and TPC2 from other mammalian species. We have tested human TPC2 with GFP or mCherry tag at either N- or C-terminus. Neither the localization nor activity was obviously changed. There are now also ample studies from multiple labs showing that the tagged TPC1 and TPC2 have similar subcellular localization and function as untagged or endogenous TPC1 and TPC2, respectively [see e.g., Wang et al., 2012; Cang et al., 2013; Ruas et al., 2015]. However, for human TPC1, the addition of an epitope tag at the N-terminus resulted in loss of protein expression (our unpublished observation). Therefore, for the current study, we only used C-terminal tagged constructs.